# Breaking the Activation Function Bottleneck through Adaptive Parameterization

**Sebastian Flennerhag**[1,2]   **Hujun Yin**[1,2]   **John Keane**[1]   **Mark Elliot**[1]

[1]University of Manchester      [2]The Alan Turing Institute

sflennerhag@turing.ac.uk {hujun.yin, john.keane, mark.elliot}@manchester.ac.uk

## Abstract

Standard neural network architectures are non-linear only by virtue of a simple element-wise activation function, making them both brittle and excessively large. In this paper, we consider methods for making the feed-forward layer more flexible while preserving its basic structure. We develop simple drop-in replacements that learn to adapt their parameterization conditional on the input, thereby increasing statistical efficiency significantly. We present an adaptive LSTM that advances the state of the art for the Penn Treebank and WikiText-2 word-modeling tasks while using fewer parameters and converging in less than half the number of iterations.

## 1  Introduction

While a two-layer feed-forward neural network is sufficient to approximate any function (Cybenko, 1989; Hornik, 1991), in practice much deeper networks are necessary to learn a good approximation to a complex function. In fact, a network tends to generalize better the larger it is, often to the point of having more parameters than there are data points in the training set (Canziani et al., 2016; Novak et al., 2018; Frankle & Carbin, 2018).

One reason why neural networks are so large is that they bias towards linear behavior: if the activation function is largely linear, so will the hidden layer be. Common activation functions, such as the Sigmoid, Tanh, and ReLU all behave close to linear over large ranges of their domain. Consequently, for a randomly sampled input to break linearity, layers must be wide and the network deep to ensure some elements lie in non-linear regions of the activation function. To overcome the bias towards linear behavior, more sophisticated activation functions have been designed (Clevert et al., 2015; He et al., 2015; Klambauer et al., 2017; Dauphin et al., 2017). However, these still limit all non-linearity to sit in the activation function.

We instead propose *adaptive parameterization*, a method for learning to adapt the parameters of the affine transformation to a given input. In particular, we present a generic adaptive feed-forward layer that retains the basic structure of the standard feed-forward layer while significantly increasing the capacity to model non-linear patterns. We develop specific instances of adaptive parameterization that can be trained end-to-end jointly with the network using standard backpropagation, are simple to implement, and run at minimal additional cost.

Empirically, we find that adaptive parameterization can learn non-linear patterns where a non-adaptive baseline fails, or outperform the baseline using 30–50% fewer parameters. In particular, we develop an adaptive version of the Long Short-Term Memory model (LSTM; Hochreiter & Schmidhuber, 1997; Gers et al., 2000) that enjoys both faster convergence and greater statistical efficiency.

The adaptive LSTM advances the state of the art for the Penn Treebank and WikiText-2 word modeling tasks using ~20–30% fewer parameters and converging in less than half as many iterations.[1] We

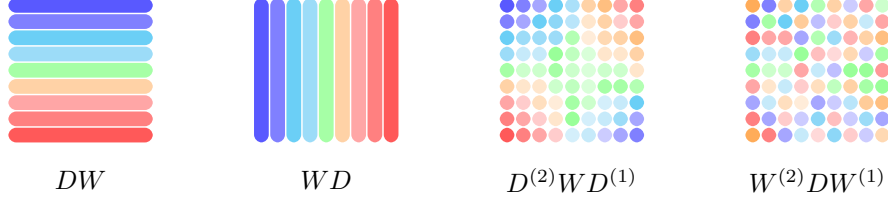

$$DW \qquad WD \qquad D^{(2)}WD^{(1)} \qquad W^{(2)}DW^{(1)}$$

Figure 1: Adaptation policies. *Left:* output adaptation shifts the mean of each row in $W$; *center left:* input adaptation shifts the mean of each column; *center right:* IO-adaptation shifts mean and variance across sub-matrices; *Right:* SVA scales singular values.

proceed as follows: section 2 presents the adaptive feed-forward layer, section 3 develops the adaptive LSTM, section 4 discusses related work and section 5 presents empirical analysis and results.

## 2 Adaptive Parameterization

To motivate adaptive parameterization, we show that deep neural networks learn a family of compositions of linear maps and because the activation function is static, the inherent flexibility in this family is weak. Adaptive parameterization is a means of increasing this flexibility and thereby increasing the model's capacity to learn non-linear patterns. We focus on the feed-forward layer, $f(\mathbf{x}) := \phi(W\mathbf{x} + \mathbf{b})$, for some activation function $\phi : \mathbb{R} \mapsto \mathbb{R}$. Define the pre-activation layer as $\mathbf{a} = A(\mathbf{x}) := W\mathbf{x} + \mathbf{b}$ and denote by $g(\mathbf{a}) := \phi(\mathbf{a})/\mathbf{a}$ the activation effect of $\phi$ given $\mathbf{a}$, where division is element-wise. Let $G = \mathrm{diag}(g(\mathbf{a}))$. We then have $f(\mathbf{x}) = g(\mathbf{a}) \odot \mathbf{a} = G\,\mathbf{a}$; we use "$\odot$" to denote the Hadamard product.[2]

For any pair $(\mathbf{x}, \mathbf{y}) \in \mathbb{R}^n \times \mathbb{R}^k$, a deep feed-forward network with $N \in \mathbb{N}$ layers, $f^{(N)} \circ \cdots \circ f^{(1)}$, approximates the relationship $\mathbf{x} \mapsto \mathbf{y}$ by a composition of linear maps. To see this, note that $\mathbf{x}$ is sufficient to determine all activation effects $\mathcal{G} = \{G^{(l)}\}_{l=1}^N$. Together with fixed transformations $\mathcal{A} = \{A^{(l)}\}_{l=1}^N$, the network can be expressed as

$$\hat{\mathbf{y}} = (f^{(N)} \circ \cdots \circ f^{(1)})(\mathbf{x}) = (G^{(N)} \circ A^{(N)} \circ \cdots \circ G^{(1)} \circ A^{(1)})(\mathbf{x}). \tag{1}$$

A neural network can therefore be understood as learning a "prior" $\mathcal{A}$ in parameter space around which it constructs a family of compositions of linear maps (as $\mathcal{G}$ varies across inputs). The neural network adapts to inputs through the set of activation effects $\mathcal{G}$. This adaptation mechanism is weak: if $\phi$ is close to linear over the distribution of $\mathbf{a}$, as is often the case, little adaptation can occur. Moreover, because $\mathcal{G}$ does not have any learnable parameters itself, the fixed prior $\mathcal{A}$ must learn to encode both global input-invariant information as well as local contextual information. We refer to this as the *activation function bottleneck*. Adaptive parameterization breaks this bottleneck by parameterizing the adaptation mechanism in $G$, thereby circumventing these issues.

To see how the activation function bottleneck arises, note that $\phi$ is redundant whenever it is closely approximated by a linear function over some non-trivial segment of the input distribution. For these inputs, $\phi$ has no non-linear effect and such lost opportunities imply that the neural network must be made larger than necessary to fully capture non-linear patterns. For instance, both the Sigmoid and the Tanh are closely approximated around 0 by a linear function, rendering them redundant for inputs close to 0. Consequently, the network must be made deeper and its layers wider to mitigate the activation function bottleneck. In contrast, adaptive parameterization places the layer's non-linearity within the parameter matrix itself, thereby circumventing the activation function bottleneck. Further, by relaxing the element-wise non-linearity constraint imposed on the standard feed-forward layer, it can learn behaviors that would otherwise be very hard or impossible to model, such as contextual rotations and shears, and adaptive feature activation.

## 2.1 The Adaptive Feed-Forward Layer

Our goal is to break the activation function bottleneck by generalizing $\mathcal{G}$ into a parameterized adaptation policy, thereby enabling the network to specialize parameters in $\mathcal{A}$ to encode global, input invariant information while parameters in $\mathcal{G}$ encode local, contextual information.

Consider the standard feed-forward layer, defined by one adaptation block $f(\mathbf{x}) = (G \circ A)(\mathbf{x})$. As described above, we increase the capacity of the adaptation mechanism $G$ by replacing it with a parameterized adaptation mechanism $D^{(j)} := \mathrm{diag}(\pi^{(j)}(\mathbf{x}))$, where $\pi^{(j)}$ is a learnable adaptation policy. Note that $\pi^{(j)}$ can be made arbitrarily complex. In particular, even if $\pi^{(j)}$ is linear, the adaptive mechanism $D^{(j)} \mathbf{a}$ is quadratic in $\mathbf{x}$, and as such escapes the bottleneck. To ensure that the adaptive feed-forward layer has sufficient capacity, we generalize it to $q \in \mathbb{N}$ adaptation blocks,[3]

$$f(\mathbf{x}) := \phi \left( D^{(q)} W^{(q-1)} \cdots W^{(1)} D^{(1)} \mathbf{x} + D^{(0)} \mathbf{b} \right). \tag{2}$$

We refer to the number of adaptation blocks $q$ as the order of the layer. Strictly speaking, the adaptive feed-forward layer does not need an activation function, but it can provide desirable properties depending on the application. It is worth noting that the adaptive feed-forward layer places no restrictions on the form of the adaptation policy $\pi = (\pi^{(0)}, \dots, \pi^{(q)})$ or its training procedure. In this paper, we parameterize $\pi$ as a neural network trained jointly with the main model. Next, we show how different adaptive feed-forward layers are generated by the choice of adaptation policy.

## 2.2 Adaptation Policies

Higher-order adaptation (i.e. $q$ large) enables expressive adaptation policies, but because the adaptation policy depends on $\mathbf{x}$, high-order layers are less efficient than a stack of low-order layers. We find that low-order layers are surprisingly powerful, and present a policy of order 2 that can express any other adaptation policy.

**Partial Adaptation**    The simplest adaptation policy ($q = 1$) is given by $f(\mathbf{x}) = WD^{(1)}\mathbf{x} + D^{(0)}\mathbf{b}$. This policy is equivalent to a mean shift and a re-scaling of the columns of $W$, or alternatively re-scaling the input. It can be thought of as a learned contextualized standardization mechanism that conditions the effect on the specific input. As such, we refer to this policy as *input adaptation*. Its mirror image, *output adaptation*, is given by $f(\mathbf{x}) = D^{(1)}W\mathbf{x} + D^{(0)}\mathbf{b}$. This is a special case of second-order adaptation policies, where $D^{(1)} = I$, where $I$ denotes the identity matrix. Both these policies are restrictive in that they only operate on either the rows or the columns of $W$ (fig. 1).

**IO-adaptation**    The general form of second-order adaptation policies integrates input- and output-adaptation into a jointly learned adaptation policy. As such we refer to this as IO-*adaptation*,

$$f(\mathbf{x}) = D^{(2)}WD^{(1)}\mathbf{x} + D^{(0)}\mathbf{b}. \tag{3}$$

IO-adaptation is much more powerful than either input- or output-adaptation alone, which can be seen by the fact that it essentially learns to identify and adapt sub-matrices in $W$ by sharing adaptation vectors across rows and columns (fig. 1). In fact, assuming $\pi$ is sufficiently powerful, IO-adaptation can express any mapping from input to parameter space.

**Property 1.** *Let $W$ be given and fix $\mathbf{x}$. For any $G$ of same dimensionality as $W$, there are arbitrarily many $(D^{(1)}, D^{(2)})$ such that $G\mathbf{x} = D^{(2)}WD^{(1)}\mathbf{x}$.*

Proof: see supplementary material.

**Singular Value Adaptation (SVA)**    Another policy of interest arises as a special case of third-order adaptation policies, where $D^{(1)} = I$ as before. The resulting policy,

$$f(\mathbf{x}) = W^{(2)} D W^{(1)} \mathbf{x} + D^{(0)} \mathbf{b}, \tag{4}$$

is reminiscent of Singular Value Decomposition. However, rather than being a decomposition, it composes a projection by adapting singular values to the input. In particular, letting $W^{(1)} = V^T A$ and $W^{(2)} = BU$, with $U$ and $V$ appropriately orthogonal, eq. 4 can be written as $B(UDV^T)A\,\mathbf{x}$, with $UDV^T$ adapted to $\mathbf{x}$ through its singular values. In our experiments, we initialize weight matrices as semi-orthogonal (Saxe et al., 2013), but we do not enforce orthogonality after initialization.

The drawback of SVA is that it requires learning two separate matrices of relatively high rank. For problems where the dimensionality of $\mathbf{x}$ is large, the dimensionality of the adaptation space has to be made small to control parameter count. This limits the model's capacity by enforcing a low-rank factorization, which also tends to impact training negatively (Denil et al., 2013).

SVA and IO-adaptation are simple but flexible policies that can be used as drop-in replacements for any feed-forward layer. Because they are differentiable, they can be trained using standard backpropagation. Next, we demonstrate adaptive parameterization in the context of Recurrent Neural Networks (RNNs), where feed-forward layers are predominant.

## 3    Adaptive Parameterization in RNNs

RNNs are common in sequence learning, where the input is a sequence $\{\mathbf{x}_1, \ldots, \mathbf{x}_t\}$ and the target variable either itself a sequence or a single point or vector. In either case, the computational graph of an RNN, when unrolled over time, will be of the form in eq. 1, making it a prime candidate for adaptive parameterization. Moreover, in sequence-to-sequence learning, the model estimates a conditional distribution $p(\mathbf{y}_t \mid \mathbf{x}_1, \ldots, \mathbf{x}_t)$ that changes significantly from one time step to the next. Because of this variance, an RNN must be very flexible to model the conditional distribution. By embedding adaptive parameterization, we can increase flexibility for a given model size. Consider the LSTM model (Hochreiter & Schmidhuber, 1997; Gers et al., 2000), defined by the gating mechanism

$$\begin{aligned} \mathbf{c}_t &= \sigma(\mathbf{u}_t^f) \odot \mathbf{c}_{t-1} + \sigma(\mathbf{u}_t^i) \odot \tau(\mathbf{u}_t^z) \\ \mathbf{h}_t &= \sigma(\mathbf{u}_t^o) \odot \tau(\mathbf{c}_t), \end{aligned} \tag{5}$$

where $\sigma$ and $\tau$ represent Sigmoid and Tanh activation functions respectively and each $\mathbf{u}_t^{s \in \{i,f,o,z\}}$ is a linear transformation of the form $\mathbf{u}_t^s = W^{(s)} \mathbf{x}_t + V^{(s)} \mathbf{h}_{t-1} + \mathbf{b}^{(s)}$. Adaptation in the LSTM can be derived directly from the adaptive feed-forward layer (eq. 2). We focus on IO-adaptation as this adaptation policy performed better in our experiments. For $\pi$, we use a small neural network to output a latent variable $\mathbf{z}_t$ that we map into each sub-policy with a projection $U^{(j)}$: $\pi^{(j)}(\mathbf{z}_t) = \tau\left(U^{(j)} \mathbf{z}_t\right)$. We test a static and a recurrent network as models for the latent variable,

$$\mathbf{z}_t = \text{ReLU}\left(W \mathbf{v}_t + \mathbf{b}\right), \tag{6}$$

$$\mathbf{z}_t = m(\mathbf{v}_t, \mathbf{z}_{t-1}) \tag{7}$$

where $m$ is a standard LSTM and $\mathbf{v}_t$ a summary variable of the state of the system, normally $\mathbf{v}_t = [\mathbf{x}_t \,;\, \mathbf{h}_{t-1}]$ (we use $[\cdot\,;\,\cdot]$ to denote concatenation). The potential benefit of using a recurrent model is that it is able to retain a separate memory that facilitates learning of local, sub-sequence specific patterns (Ha et al., 2017). Generally, we find that the recurrent model converges faster and generalizes marginally better. To extend the adaptive feed-forward layer to the LSTM, index sub-policies with a tuple $(s, j) \in \{i, f, o, z\} \times \{0, 1, 2, 3, 4\}$ such that $D_t^{(s,j)} = \text{diag}(\pi^{(s,j)}(\mathbf{z}_t))$. At each time step $t$ we adapt the LSTM's linear transformations through IO-adaptation,

$$\mathbf{u}_t^s = D_t^{(s,4)} W^{(s)} D_t^{(s,3)} \mathbf{x}_t + D_t^{(s,2)} V^{(s)} D_t^{(s,1)} \mathbf{h}_{t-1} + D_t^{(s,0)} \mathbf{b}^{(s)}. \tag{8}$$

An undesirable side-effect of the formulation in eq. 8 is that each linear transformation requires its own modified input, preventing a vectorized implementation of the LSTM. We avoid this by tying all input adaptations across $s$: that is, $D^{(s',j)} = D^{(s,j)}$ for all $(s',j) \in \{i, f, o, z\} \times \{1, 3\}$. Doing so approximately halves the computation time and speeds up convergence considerably. When stacking multiple aLSTM layers, the computational graph of the model becomes complex in that it extends both in the temporal dimension and along the depth of the stack. For the recurrent adaptation policy (eq. 7) to be consistent, it should be conditioned not only by the latent variable in its own layer, but also on that of the preceding layer, or it will not have a full memory of the computational graph. To achieve this, for a layer $l \in \{1, \ldots, L\}$, we define the input summary variable as

$$\mathbf{v}_t^{(l)} = \left[ \mathbf{h}_t^{(l-1)} \, ; \mathbf{h}_{t-1}^{(l)} \, ; \mathbf{z}_t^{(l-1)} \right], \tag{9}$$

where $\mathbf{h}_t^{(0)} = \mathbf{x}_t$ and $\mathbf{z}_t^{(0)} = \mathbf{z}_{t-1}^{(L)}$. In doing so, the credit assignment path of adaption policy visits all nodes in the computational graph. The resulting adaptation model becomes a blend of a standard LSTM and a Recurrent Highway Network (RHN; Zilly et al., 2016).

## 4 Related Work

Adaptive parameterization is a special case of having a relatively inexpensive learning algorithm search a vast parameter space in order to parameterize the larger main model (Stanley et al., 2009; Fernando et al., 2016). The notion of using one model to generate context-dependent parameters for another was suggested by Schmidhuber (1992); Gomez & Schmidhuber (2005). Building on this idea, Ha et al. (2017) proposed to jointly train a small network to generate the parameters of a larger network; such HyperNetworks have achieve impressive results in several domains (Suarez, 2017; Ha & Eck, 2018; Brock et al., 2018). The general concept of learning to parameterize a model has been explored in a variety of contexts, for example Schmidhuber (1992); Gomez & Schmidhuber (2005); Denil et al. (2013); Jaderberg et al. (2017); Andrychowicz et al. (2016); Yang et al. (2018).

Parameter adaptation has also been explored in meta-learning, usually in the context of few-shot learning, where a meta-learner is trained across a set of tasks to select task-specific parameters of a downstream model (Bengio et al., 1991, 1995; Schmidhuber, 1992). Similar to adaptive parameterization, Bertinetto et al. (2016) directly tasks a meta learner with predicting the weights of the task-specific learner. Ravi & Larochelle (2017) defines the adaptation policy as a gradient-descent rule, where the meta learner is an LSTM tasked with learning the update rule to use. An alternative method pre-defines the adaptation policy as gradient descent and meta-learns an initialization such that performing gradient descent on a given input from some new task yields good task-specific parameters (Finn et al., 2017; Lee & Choi, 2017; Al-Shedivat et al., 2018).

Using gradient information to adjust parameters has also been explored in sequence-to-sequence learning, where it is referred to as dynamic evaluation (Mikolov, 2012; Graves, 2013; Krause et al., 2017). This form of adaptation relies on the auto-regressive property of RNNs to adapt parameters at each time step by taking a gradient step with respect to one or several previous time steps.

Many extensions have been proposed to the basic RNN and the LSTM model (Hochreiter & Schmidhuber, 1997; Gers et al., 2000), some of which can be seen as implementing a form of constrained adaptation policy. The multiplicative RNN (mRNN; Sutskever et al., 2011) and the multiplicative LSTM (mLSTM; Krause et al., 2016) can be seen as implementing an SVA policy for the hidden-to-hidden projections. mRNN improves upon RNNs in language modeling tasks (Sutskever et al., 2011; Mikolov et al., 2012), but tends to perform worse than the standard LSTM (Cooijmans et al., 2016). mLSTM has been shown to improve upon RNNs and LSTMs on language modeling tasks (Krause et al., 2017; Radford et al., 2017). The multiplicative-integration RNN and its LSTM version (Wu et al., 2016) essentially implement a constrained output-adaptation policy.

The implicit policies in the above models conditions only on the input, ignoring the state of the system. In contrast, the GRU (Cho et al., 2014; Chung et al., 2014) can be interpreted as implementing an input-adaptation policy on the input-to-hidden matrix that conditions on both the input and the state of the system. Most closely related to the aLSTM are HyperNetworks (Ha et al., 2017; Suarez, 2017); these implement output adaptation conditioned on both the input and the state of the system using a recurrent adaptation policy. HyperNetworks have attained impressive results on character level

modeling tasks and sequence generation tasks, including hand-writing and drawing sketches (Ha et al., 2017; Ha & Eck, 2018). They have also been used in neural architecture search by generating weights conditional on the architecture (Brock et al., 2018), demonstrating that adaptive parameterization can be conditioned on some arbitrary context, in this case the architecture itself.

# 5 Experiments

We compare the behavior of a model with adaptive feed-forward layers to standard feed-forward baselines in a controlled regression problem and on MNIST (LeCun et al., 1998). The aLSTM is tested on the Penn Treebank and WikiText-2 word modeling tasks. We use the ADAM optimizer (Kingma & Ba, 2015) unless otherwise stated.

## 5.1 Extreme Tail Regression

To study the flexibility of the adaptive feed-forward layer, we sample $\mathbf{x} = (x_1, x_2)$ from $\mathcal{N}(\mathbf{0}, I)$ and construct the target variable as $y = (2x_1)^2 - (3x_2)^4 + \epsilon$ with $\epsilon \sim \mathcal{N}(0, 1)$. Most of the data lies on a hyperplane, but the target variable grows or shrinks exponentially as $x_1$ or $x_2$ moves away from 0. We compare a 3-layer feed-forward network with 10 hidden units to a 2-layer model with 2 hidden units, where the first layer is adaptive and the final layer is static. We use an SVA policy where $\pi$ is a gated linear unit (Dauphin et al., 2017). Both models are trained for 10 000 steps with a batch size of 50 and a learning rate of 0.003.

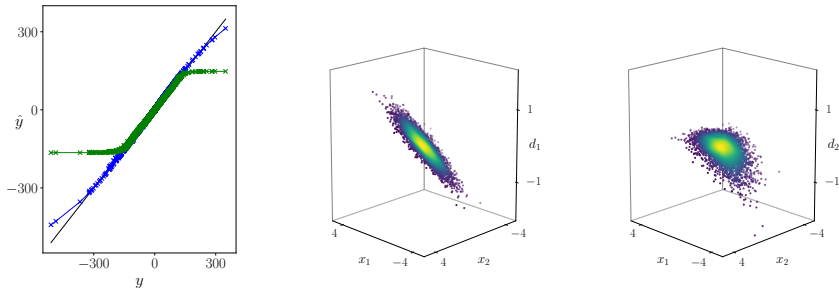

Figure 2: Extreme tail regression. *Left:* Predictions of the adaptive model (blue) and the baseline model (green) against ground truth (black). *Center & Right:* distribution of adaptive singular values.

The baseline model fails to represent the tail of the distribution despite being three times larger. In contrast, the adaptive model does a remarkably good job given how small the model is and the extremity of the distribution. It is worth noting how the adaptation policy encodes local information through the distribution of its singular values (fig. 2).

## 5.2 MNIST

We compare performance of a 3-layer feed-forward model against (a) a single-layer SVA model and (b) a 3-layer SVA model. We train all models with Stochastic Gradient Descent with a learning rate of 0.001, a batch size of 128, and train for 50 000 steps. The single-layer adaptive model reduces to a logistic regression conditional on the input. By comparing it to a logistic regression, we measure the marginal benefit of the SVA policy to approximately 1 percentage point gain in accuracy. In fact, if the one-layer SVA model has a sufficiently expressive adaptation model it matches and even outperforms the deep feed-forward baseline.

## 5.3 Penn Treebank

The Penn Treebank corpus (PTB; Marcus et al., 1993; Mikolov et al., 2010) is a widely used benchmark for language modeling. It consists of heavily processed news articles and contains no capital letters, numbers, or punctuation. As such, the vocabulary is relatively small at 10 000 unique words. We evaluate the aLSTM on word-level modeling following standard practice in training setup (e.g. Zaremba et al., 2015). As we are interested in statistical efficiency, we fix the number of layers to 2,

Table 1: Train and test set accuracy on MNIST

| Model | Size | Train | Test |
|---|---|---|---|
| Logistic Regression | 8K | 92.00% | 92.14% |
| 3-layer feed-forward | 100K | 97.57% | 97.01% |
| 1-layer SVA | 8K | 94.05% | 93.86% |
| 1-layer SVA | 100K | 98.62% | 97.14% |
| 3-layer SVA | 100K | **99.99%** | **97.65%** |

though more layers tend to perform better, and use a policy latent variable size of 100. For details on hyper-parameters, see supplementary material. As we are evaluating underlying architectures, we do not compare against bolt-on methods (Grave et al., 2017; Yang et al., 2018; Mikolov, 2012; Graves, 2013; Krause et al., 2017). These are equally applicable to the aLSTM.

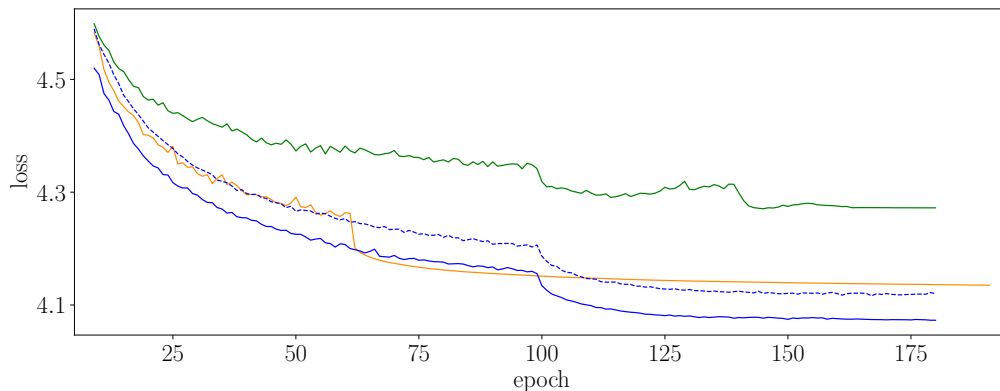

Figure 3: Validation loss on PTB for our LSTM (green), aLSTM (blue), aLSTM with static policy (dashed), and the AWD-LSTM (orange; Merity et al., 2018). Drops correspond to learning rate cuts.

The aLSTM improves upon previously published results using roughly 30% fewer parameters, a smaller hidden state size, and fewer layers while converging in fewer iterations (table 2). Notably, for the standard LSTM to converge at all, gradient clipping is required and dropout rates must be reduced by ~25%. In our experimental setup, a percentage point change to these rates cause either severe overfitting or failure to converge. Taken together, this indicates that adaptive parameterization enjoys both superior stability properties and substantially increases model capacity, even when the baseline model is complex; we explore both further in sections sections 5.5 and 5.6. Melis et al. (2018) applies a large-scale hyper-parameter search to an LSTM version with tied input and forget gates and inter-layer skip-connections (TG-SC LSTM), making it a challenging baseline that the aLSTM improves upon by a considerable margin.

Previous state-of-art performance was achieved by the ASGD Weight-Dropped LSTM (AWD-LSTM; Merity et al., 2018), which uses regularization, optimization, and fine-tuning techniques designed specifically for language modeling[4]. The AWD-LSTM requires approximately 500 epochs to converge to optimal performance; the aLSTM outperforms the AWD-LSTM after 144 epochs and converges to optimal performance in 180 epochs. Consequently, even if the AWD-LSTM runs on top of the CuDNN implementation of the LSTM, the aLSTM converges approximately ~25% faster in wall-clock time. In summary, any form of adaptation is beneficial, and a recurrent adaptation model (eq. 7) enjoys both fastest convergence rate and best final performance in this experiment.

Table 2: Validation and test set perplexities on Penn Treebank. All results except those from Zaremba et al. (2015) use tied input and output embeddings (Press & Wolf, 2017).

| Model | Size | Depth | Valid | Test |
|---|---|---|---|---|
| LSTM, Zaremba et al. (2015) | 24M | 2 | 82.2 | 78.4 |
| RHN, Zilly et al. (2016) | 24M | 10 | 67.9 | 65.4 |
| NAS, Zoph & Le (2017) | 54M | — | — | 62.4 |
| TG-SC LSTM, Melis et al. (2018) | 10M | 4 | 62.4 | 60.1 |
| TG-SC LSTM, Melis et al. (2018) | 24M | 4 | 60.9 | 58.3 |
| AWD-LSTM, Merity et al. (2018) | 24M | 3 | 60.0 | 57.3 |
| LSTM | 20M | 2 | 71.7 | 68.9 |
| aLSTM, static policy (eq. 6) | 17M | 2 | 60.2 | 58.0 |
| aLSTM, recurrent policy (eq. 7) | 14M | 2 | 59.6 | 57.2 |
| aLSTM, recurrent policy (eq. 7) | 17M | 2 | 58.7 | 56.5 |
| aLSTM, recurrent policy (eq. 7) | 24M | 2 | **57.6** | **55.3** |

## 5.4 WikiText-2

WikiText-2 (WT2; Merity et al., 2017) is a corpus curated from Wikipedia articles with lighter processing than PTB. It is about twice as large with three times as many unique tokens. We evaluate the aLSTM using the same settings as on PTB, and additionally test a version with larger hidden state size to match the parameter count of current state of the art models. Without tuning for WT2, both outperform previously published results in 150 epochs (table 3) and converge to new state of the art performance in 190 epochs. In contrast, the AWD-LSTM requires 700 epochs to reach optimal performance. As such, the aLSTM trains ~40% faster in wall-clock time. The TG-SC LSTM in Melis et al. (2018) uses fewer parameters, but its hyper-parameters are tuned for WT2, in contrast to both the AWD-LSTM and aLSTM. We expect that tuning hyper-parameters specifically for WT2 would yield further gains.

Table 3: Validation and test set perplexities on WikiText-2.

| Model | Size | Depth | Valid | Test |
|---|---|---|---|---|
| LSTM, Grave et al. (2017) | — | — | — | 99.3 |
| LSTM, Inan et al. (2017) | 22M | 3 | 91.5 | 87.7 |
| AWD-LSTM, Merity et al. (2018) | 33M | 3 | 68.6 | 65.8 |
| TG-SC LSTM, Melis et al. (2018) | 24M | 2 | 69.1 | 65.9 |
| aLSTM, recurrent policy (eq. 7) | 27M | 2 | 68.1 | 65.5 |
| aLSTM, recurrent policy (eq. 7) | 32M | 2 | **67.5** | **64.5** |

## 5.5 Ablation Study

We isolate the effect of each component in the aLSTM through an ablation study on PTB. We adjust the hidden state to ensure every model has approximately 17M learnable parameters. We use the same hyper-parameters for all models except for (a) the standard LSTM (see above) and (b) the aLSTM under an output-adaptation policy and a feed-forward adaptation model, as this configuration needed slightly lower dropout rates to converge to good performance.

As table 4 shows, any form of adaptation yields a significant performance gain. Going from a feed-forward adaptation model (eq. 6) to a recurrent adaptation model (eq. 7) yields a significant improvement irrespective of policy, and our hybrid RHN-LSTM (eq. 9) provides a further boost. Similarly, moving from a partial adaptation policy to IO-adaptation leads to significant performance improvement under any adaptation model. These results indicate that the LSTM is constrained by the activation function bottleneck and increasing its adaptive capacity breaks the bottleneck.

Table 4: Ablation study: perplexities on Penn Treebank.[†]Equivalent to the HyperNetwork, except the aLSTM uses one projection from $\mathbf{z}$ to $\pi$ instead of nesting two (Ha et al., 2017).

| Model | Adaptation model | Adaptation policy | Valid | Test |
|---|---|---|---|---|
| LSTM | — | — | 71.7 | 68.9 |
| aLSTM | feed-forward | output-adaptation | 66.0 | 63.1 |
| aLSTM[†] | LSTM | output-adaptation | 59.9 | 58.2 |
| aLSTM | LSTM-RHN | output-adaptation | 59.7 | 57.3 |
| aLSTM | feed-forward | IO-adaptation | 61.6 | 59.1 |
| aLSTM | LSTM | IO-adaptation | 59.0 | 56.9 |
| aLSTM | LSTM-RHN | IO-adaptation | **58.5** | **56.5** |

## 5.6 Robustness

We further study the robustness of the aLSTM with respect to hyper-parameters. We limit ourselves to dropout rates and train for 10 epochs on PTB. All other hyper-parameters are held fixed. For each model, we draw 100 random samples uniformly from intervals of the form $[r - 0.1, r + 0.1]$, with $r$ being the optimal rate found through previous hyper-parameter tuning. The two models exhibit very different distributions (fig. 4). The distribution of the aLSTM is tight, reflecting robustness with respect to hyper-parameters. In fact, no sampled model fails to converge. In contrast, approximately 25% of the population of LSTM configurations fail to converge. In fact, fully 45% of the LSTM population fail to outperform the *worst* aLSTM configuration; the 90[th] percentile of the aLSTM distribution is on the same level as the 10[th] percentile of the LSTM distribution. On WT-2 these results are amplified, with half of the LSTM population failing to converge and 80% of the LSTM population failing to outperform the worst-case aLSTM configuration.

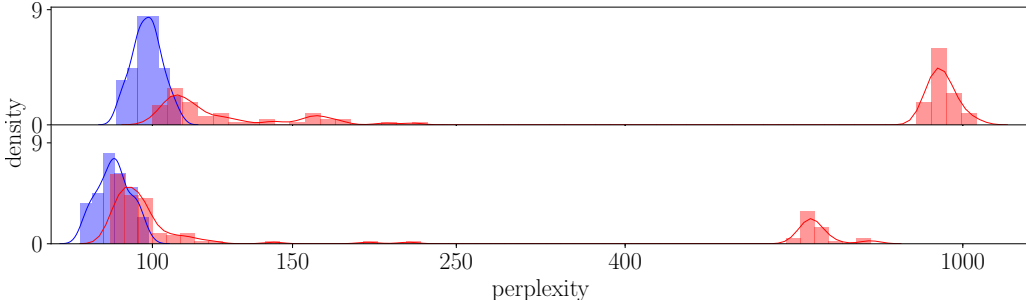

Figure 4: Distribution of validation scores on WikiText-2 (top) and Penn Treebank (bottom) for randomly sampled hyper-parameters. The aLSTM (blue) is more robust than the LSTM (red).

## 6 Conclusions

By viewing deep neural networks as adaptive compositions of linear maps, we have showed that standard activation functions induce an activation function bottleneck because they fail to have significant non-linear effect on a non-trivial subset of inputs. We break this bottleneck through adaptive parameterization, which allows the model to adapt the affine transformation to the input.

We have developed an adaptive feed-forward layer and showed empirically that it can learn patterns where a deep feed-forward network fails whilst also using fewer parameters. Extending the adaptive feed-forward layer to RNNs, we presented an adaptive LSTM that significantly increases model capacity and statistical efficiency while being more robust to hyper-parameters. In particular, we obtain new state of the art results on the Penn Treebank and the WikiText-2 word-modeling tasks, using ~20–30% fewer parameters and converging in less than half as many iterations.

**Acknowledgments**

The authors would like to thank anonymous reviewers for their comments. This work was supported by ESRC via the North West Doctoral Training Centre, grant number ES/J500094/1.

## Footnotes

[1]Code available at https://github.com/flennerhag/alstm.

[2]This holds almost everywhere, but not for $\{\mathbf{a} \mid a_i = 0, a_i \in \mathbf{a}\}$. Being measure 0, we ignore this exception.

[3]The ordering of $W$ and $D$ matrices can be reversed by setting the first and / or last adaptation matrix to be the identity matrix.

[4]Public release of their code at `https://github.com/salesforce/awd-lstm-lm`

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
