[Supplementary Material]

## Supplementary Material

## A    Proof of property 1

Property 1 asserts that for given $W$ and $\mathbf{x}$, for any $G$ of same dimensionality as $W$, there are arbitrarily many $(D^{(1)}, D^{(2)})$ such that $G\mathbf{x} = D^{(2)}WD^{(1)}\mathbf{x}$. We now prove this property:

*Proof.* Let $\mathbf{y}^G = G\mathbf{x} = \sum_i x_i\mathbf{g}_i$, where $\mathbf{g}_i$ is the $i$th column of $G$. Let $d_i^{(s)} = D_{ii}^{(s)}$ and write

$$D^{(2)}WD^{(1)}\mathbf{x} = \sum_i x_i d_i^{(1)} \begin{bmatrix} d_1^{(2)}w_{1i} \\ \vdots \\ d_m^{(2)}w_{mi} \end{bmatrix}. \tag{10}$$

Consequently, choose some $k$ for which $x_k \neq 0$ and set $d_k^{(1)} = 1/x_k$, $d_j^{(2)} = y_j^G/w_{jk}$. We have

$$D^{(2)}WD^{(1)}\mathbf{x} = \frac{x_k}{x_k} \begin{bmatrix} (y_1^G/w_{1k})w_{1k} \\ \vdots \\ (y_m^G/w_{mk})w_{mk} \end{bmatrix} = \mathbf{y}^G. \tag{11}$$

Moreover, since this holds for any $x_k \neq 0$, every linear combination of such solutions are also solutions. ∎

## B    Hyper-parameters for PTB and WT2

We use tied embedding weights (Press & Wolf, 2017; Inan et al., 2017) and a weight decay rate of $10^{-6}$. The initial learning rate is set to 0.003 with decay rates $\beta_1 = 0$ and $\beta_2 = 0.999$. The learning rate is cut after epochs 100 and 160 by a factor of 10. We use a batch size of 20 and variable truncated backprogagation through time centered at 70, as in Merity et al. (2018). Dropout rates were tuned with a coarse random search. We apply variational dropout (Gal & Ghahramani, 2016) to word and word embedding vectors, the policy latent variable, the hidden state of the aLSTM and the final aLSTM output with drop probabilities 0.16, 0.6, 0.1, 0.25, and 0.6 respectively.