[Reviews · NeurIPS 2018]

Reviewer 1



The authors propose Adaptive Parameterization, a network architecture that largely replaces nonlinear activation functions with learnable input-dependent transformations. The parameters of these transformations are proposed to be produced by another neural network. The authors describe several examples of this architecture and present results on standard benchmarks using an RNN whose recurrent function implements the proposed Adaptive Parameterization. The work is clearly presented and well motivated, and the results are compelling. As far as I know, the core concept is original, although indeed closely related to some previous work, as noted by the authors. The work could be strengthened by streamlining the mathematical notation and conceptual flow in sections 2 and 3. The concepts presented here are rather simple, and readers would benefit if the authors clarified this presentation as possible (e.g., there seem to be many auxiliary variables and functions floating around--eliminating these might reduce the working memory burden placed on readers). Further, some of the links made to the "Example Policies" in section 2.2 are rather imprecise and might be worth striking or rephrasing (e.g.., "partial adaptation" only standardizes inputs / outputs in the unlikely event that the parameters learned end up z-scoring the inputs or outputs; "singular value adaptation" only relates to SVD in the unlikely event that the learned parameters have the appropriate orthogonal structure).

Reviewer 2



In this work, the authors propose to replace the static non-linear activation functions used in most modern neural networks with context-dependent linear transformations. The authors test their model on a toy dataset and MNIST digit classification as well as on two language modeling tasks, PTB and Wikitext-2. On the latter, they show that their approach out-performs recent comparable architectures. The paper is mostly clearly written and easy to follow, and the results support the authors' claim (still, running language modeling experiments on datasets at least an order of magnitude larger would significantly strengthen the paper). Further discussion of the added computational cost of the method, which can be as important as the number of parameters both for practical applications and in measuring a model's expressive power, would also be welcome. A few other random points: - The first part of Section 2 seems to initially motivate the approach as a first order approximation of the regular non-linear activation functions: if that is indeed its purpose, it might help to make it more explicit. - In Figure 3, it looks like the learning rate is cut after 60 epochs on AWD-LSTM and 100 for aLSTM, why is that? - Also, Figure 3 would be more legible with a legend in the graph.

Reviewer 3



The work proposes a form of adaptive parameterization for the activation function in neural nets. Several instantiations for feed-forward layers and a delicate adaptive design for LSTM are presented. Empirically, the adaptive version with few parameters outperforms the static version for MNIST classification and word-level language modeling. While the idea of using adaptive parameterization is not new, the work introduces the concept of activation function bottleneck, which is not rigorously defined. If I understand correctly, the bottleneck conceptually refers to the fact that since the non-linearity is a static function of the pre-activation, in order to well exploit the "non-linear scaling" effect of the non-linearity, the weight matrix has to additionally encode some input specific information. As a result, the capacity of the weight matrix is not used efficiently. To break the bottleneck, one can make the weight matrix input specific (HyperNetworks) or the entire layer input specific in a more complicated way. This work proposes a particular input-dependent parameterization of a layer by (1) removing the nonlinearity and (2) interleaving adaptive diagonal matrices produced by hyper-networks and static weight matrices. As the parameterization has good empirical performance, there are a few issues: - The proof of Property 1 is not satisfying, as it essentially assumes pi(x) can approximate G x / W. - Further, while the original nonlinearity in a layer is removed, there are nonlinearities in the HyperNetworks. Hence, I tend to feel that the proposed adaptive layer is more like a more complicated sub-network with "multiplicative interactions" between various components, where some of these components are affine mappings of the input and others are non-linear mappings of the input. Thus, I'm not quite convinced by the perspective that the proposed layer structure is just an adaptive parameterization of the activation function. In addition, for the Penn Treebank experiment, the performance improvement of the aLSTM with 17M parameters is relatively small compared to the improvement of the aLSTM with comparable parameter size in the wikitext-2 experiment. So, it's better to run an aLSTM with comparable parameter size to AWD-LSTM for Penn Treebank, which is more informative. In summary, this work hypothesizes a "capacity bottlenck" due to the static behavior of the activation function without a formal definition. Notably, an effective adaptive parameterization of LSTM is proposed. However, no detailed analysis is given to justify the effectiveness of the proposed method, leaving the source of the effectiveness not clear.